



# IITM High-Resolution Global Forecast Model Version 1: An attempt to resolve monsoon prediction deadlock

R. Phani Murali Krishna[1], Siddharth Kumar[1], A. Gopinathan Prajeesh[2], Peter Bechtold[3], Nils Wedi[3], Kumar Roy[4], Malay Ganai[1], B. Revanth Reddy[1], Snehlata Tirkey[1], Tanmoy Goswami[1] , Radhika Kanase[1], and Parthasarathi Mukhopadhyay[1]

[1]Indian Institute of Tropical Meteorology, Ministry of Earth Sciences, Dr. Homi Bhabha Road, Pune 411008, India
[2] King Abdullah University of Science and Technology, Saudi Arabia
[3] ECMWF
[4] University of Victoria, Canada

Correspondence to: Dr. P. Mukhopadhyay (mpartha@tropmet.res.in; parthasarathi64@gmail.com)



**Abstract.** The prediction of Indian monsoon rainfall variability affecting a country with a population of billions remained
one of the major challenges of the numerical weather prediction community. While in recent years, there has been a
significant improvement in predicting the synoptic scale transients associated with the monsoon circulation, the intricacies of
rainfall variability remained a challenge. Here, an attempt is made to develop a global model using a dynamic core of a cubic
octahedral grid that provides a higher resolution of 6.5 km over the global tropics. This high-resolution model has been
developed to resolve the monsoon convection. Reforecasts with the IITM High-resolution Global Forecast Model (HGFM)
have been run daily from June through September 2022. The HGFM model has a wave number truncation of 1534 in the
cubic octahedral grid. The monsoon events have been predicted with a ten-day lead time. The HGFM model is compared to
the operational GFS T1534. While the HGFM provides skills comparable to the GFS, it shows better skills for higher
precipitation thresholds. This model is currently being run in experimental mode and will be made operational.





## 1 Introduction

In spite of significant improvement in numerical weather prediction skill in the last decades (Bechtold et al., 2008; Magnusson and Kallen 2013; Hoffman et al., 2018) predictions of tropical rainfall variability remain a challenge (Westra et al., 2014; Prakash et al., 2016). Stephens et al. (2010) demonstrated that the models predict in the tropics too many rainy days which are in the lighter rain category. The challenges of tropical rainfall variability have also been demonstrated by Watson et al., 2017. The vagaries of the Indian monsoon every year affect the lifestyle of billions of people and the economy of the Indian sub-continent modulating its Gross Domestic Product (GDP) (Gadgil and Gadgil, 2006). It is therefore of the utmost importance to improve the weather prediction skill in general and extreme precipitation events in particular. With the increase of computing power, the resolution of numerical weather prediction models have been increasing and global models with a resolution of 1~7 km have become a reality (Miura et al., 2007; Satoh et al., 2005; Satoh et al., 2019; Wedi et al., 2020). The higher resolution of Numerical Weather Prediction (NWP) models has been found to produce a realistic rainfall variability across scales including diurnal variation, better Madden Julian Oscillation (MJO) variability and seasonal mean climate (Kinter et al., 2013; Rajendran and Kitoh, 2008; Skamarock et al., 2012; Molod et al., 2015; Crueger et al. 2018; Giorgetta et al., 2018). In India, operational NWP was initiated with moderate resolution of T80 and then gradually enhanced to T382, T574 (Prasad et al., 2011, 2014, 2017) and very recently to T1534 (Mukhopadhyay et al., 2019). The advantage of using higher resolution (T1534~12.5 km) as against the lower resolution T574 (~27 km) was found by enhancement of the model skill by 2 days (Rao et al., 2019). The National Centre for Environmental Prediction (NCEP) GFS model with 21 members has been used for probabilistic forecasts since June 2018 (Deshpande et al., 2021). The high-resolution GFS T1534 is found to enhance the skill of heavy rainfall event (Mukhopadhyay et al., 2019), tropical cyclones and even block level prediction of rainfall (block is a sub-division of a districts in India, typically of the size of the grid of GFS T1534). However, the skill of the GFS T1534 for prediction of extremely heavy precipitation can still be improved particularly over the orographic regions of India such as the southern coastal state of Kerala, India (Mukhopadhyay et al., 2021).

The 12-km deterministic and the ensemble model based on the GFS do show reasonably good skill in capturing the monsoon rainfall with 3 to 5 days lead time. The skill of the GFS forecast for Indian monsoon has been reported by Mukhopadhyay et al. (2019) and the skill of tropical cyclones with the Global Ensemble Forecast System (GEFS) has also been reported in Deshpande et al. (2021). However, in a recent study Mukhopadhyay et al. (2021) showed that three state-of-the-art ensemble forecast systems namely the GEFS, the United Kingdom Meteorological Office (UKMO) based NCMRWF Ensemble Prediction System (NEPS) run by National Centre for Medium Range Weather Forecasting (NCMRWF) and the Integrated Forecasting System (IFS) by ECMWF struggled to capture the extremely heavy rainfall over Kerala state of India during August 2018 and August 2019 extremely heavy rainfall episode. This in fact brought up the limitation of the model in resolving the rainfall variability over the Indian region and more importantly over the orographic region. One of the limitations in resolving the regional variabilities of rainfall is the horizontal resolution which does not allow the model to resolve the smaller scale processes. Therefore, a need was felt to enhance the horizontal resolution of the existing GFS based





forecasting system. As running of a model close to the convection permitting model (at a resolution lesser than 10 km) is
computationally too expensive in conventional linear reduced Gaussian grids, it was thought to build a weather model with a
grid which has a variable resolution from the pole to the equator. In view of this, the Tco has been identified and the GFS
linear reduced Gaussian Grid at triangular truncation 1534 is replaced by an equivalent truncation of 1534 in cubic
octahedral grid. The equivalent model resolutions of the linear Tl1534 and the cubic Tco1543 grids are displayed in Fig. 1a.
Indeed, as the linear grid has a roughly uniform grid point resolution of 12.5 km the octahedral grid has a resolution of about
8 km in the Polar Regions and around 6 km in the tropical band. One of the prominent examples of the Global NWP model
with the Tco grid is that of the European Centre for Medium-Range Weather Forecasts (ECMWF) model suites. The Tco
grid provides several advantages (ECMWF Documentation Cy43r1, 2016) over that of the conventional reduced Gaussian
linear grid (Fig. 1a), to name a few- significant reduction in computation cost, improved representation of orography, better
filtering and better conservation properties. These properties of Tco make it a better candidate, particularly for the utilization
of high-performance computers (HPC).
This paper is the first attempt to best of our knowledge, towards building a model close to a convection permitting global
weather model in India with an emphasis to Indian monsoon rainfall variability. The details of the model development and
the experiments conducted have been elaborated in Sect. 2. The model results are analysed in Sect. 3, and the conclusion of
the study is summarized in Sect. 4.

## 2 Model, Data and Methodology

This A new grid, namely the Triangular Cubic Octahedral (Tco) grid, has been adopted to change the existing GFS (semi-
lagrangian) Gaussian linear model system. In the spectral domain, dynamical fields are represented by the sum of spherical
harmonics. The total wavenumber characterizes the spherical harmonics, and the associated wavelength is the ratio of the
circumference of the Earth to the total wavenumber. The value of the maximum wavenumber (n_max) used to represent a
field as the sum of spherical harmonics is also the spectral truncation of the model. In the case of both GFS and Tco, the
value of n_max is 1534. For the same spectral truncation n_max, the number of latitude circles from the equator to the pole
can vary depending on the choice of spectral transformation. For a linear grid, n_max=2N-1, and for a cubic grid, n_max=N-
1. Therefore, for a linear Gaussian grid, the smallest wavelength is represented by only two grid points, as is the case with
the GFS 1534 model. However, in the case of triangular truncation, the smallest wavelength is represented by four grid
points (in the case of the Tco grid). In triangular truncation, for the same spectral truncation, the number of latitude circles is
about double that of the linear truncation. For the GFS model, the horizontal resolution is ~12.5 km, and applying the cubic
grid ensures that the horizontal resolution becomes ~6.5 km in the tropics (about half of the currently used model resolution)
for the Tco grid. In the Tco grid, the number of latitude circles is 1535.
Once a particular choice of spectral truncation is made, the number of latitude circles becomes obvious. However, the
number of longitude circles per latitude circle remains to be prescribed for the creation of the global grid structure. In a fully





Gaussian grid, the number of longitude circles per latitude circle remains the same throughout the latitudes from the equator
to the pole. Thus, the effective resolution near the poles becomes very high compared to the equatorial regions. This specific
requirement demands too many computational resources and poses problems of numerical instability. To overcome that, in
the linear Gaussian grid, the number of latitude circles decreases in a certain way from the equator toward the pole to ensure
almost the same zonal resolution. For the cubic octahedral grid, the number of longitude points per latitude circle is
prescribed in a different way. The latitude circle closest to the pole consists of 20 longitude points, and the number of
longitude points increases by 4 at each latitude circle, moving from poles towards the equator. The number of longitude
points at the equator in the case of the Tco grid is given by Nx=20+1534*4=6156. Therefore, the zonal grid
length=2pi*R/Nx~6.5 km. In the original reduced Gaussian grid, the number of longitude points per latitude remains fixed in
different blocks of latitudes. The number of latitude points jumps from one block to the other by a constant number. Unlike
the linear reduced Gaussian grid, the horizontal resolution varies more smoothly with latitudes in Tco. The Collignon
projection of a sphere obtains this configuration onto an octahedron. In the current study, the Tco grid at truncation
wavenumber of 1534 is being used. Fig. 1a and Fig. 1b depicts the variation of grid resolution with latitude in the GFS (SL)
and HGFM (Tco).
Before testing the HGFM with complete physics (see Table 1 for description of physics being used in both versions of
model), we have made a version of HGFM with only a dynamical core following Held and Suarez (1994), referred to as
HS94. The HS94 is run to check the stability of the Tco grid framework. Surface boundary conditions for the Tco grid have
been meticulously prepared to ensure the accuracy of grid-point representation. Moreover, the HGFM (Tco1534) has been
developed with complete physics and incorporates essential boundary conditions, including global topography, global land-
use-land-cover etc. The HGFM at Tco1534 truncation is depicted over the globe in Fig. 1. The model has been run daily for
a ten days forecast at IITM Pratyush HPC system. To understand the computational efficiency of Tco model, time taken for
one day forecast is compared for GFS 1534 and HGFM model (Tco 765 in this case) (see Fig. 1c). A comparison between
GFS 1534 and Tco 765 is made because both models have almost same number of grid points. It is clear that Tco 765
significantly saves the runtime in dynamical core and total time as well. Moreover, Tco model is in general more scalable for
higher number of cores (not shown). The model has been run for the summer monsoon season of June, July, August and
September (JJAS) 2022. A detailed analysis of the model run has been discussed in the results section. Apart from the
monsoon season, few case studies have also been discussed.



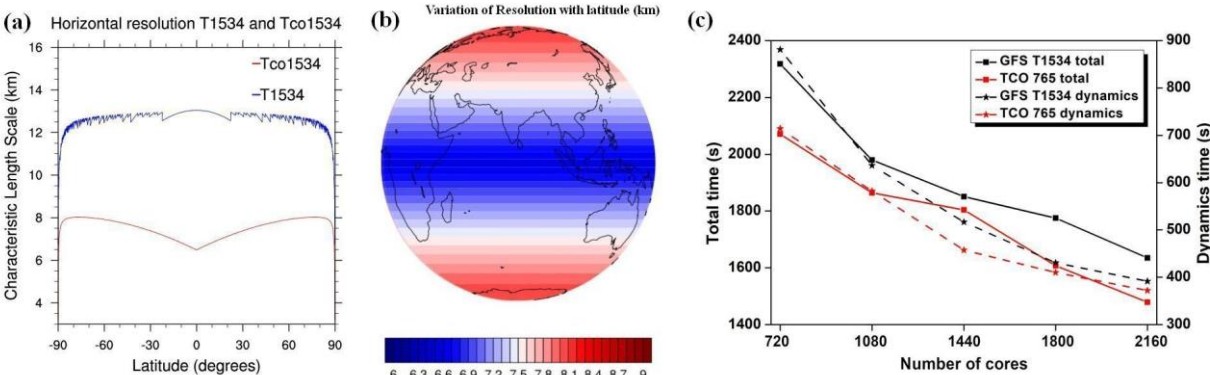

**Figure 1. Variation of grid length with latitude in GFS (blue) and Tco (red) (a), depiction of grid resolution over the globe in Tco grid (b), total and dynamics time taken for different number of cores (c). Time taken by GFS and HGFM for one day forecast (Left vertical axis is total time taken and right axis represents time taken by model dynamics).**

To verify the model forecast, the daily observed gridded rainfall data from the Integrated Multi-satellite Retrievals for GPM (IMERG) version 06B (Huffman et al., 2019) rainfall data at 0.1° × 0.1° (10 km) horizontal resolution is utilized for the year of 2022 for JJAS season. Additionally, to validate a heavy rainfall event over India, gridded rainfall from India Meteorological Department (IMD) at 25 km resolution is used. The IMD rainfall data are merged product of gridded rain gauge observations and GPM satellite-estimated rainfall over the ISM region (Mitra et al., 2014). Further, the reanalysis-based parameters from the fifth generation of ECMWF atmospheric reanalyses (ERA5) products (Hersbach and Dee, 2016) are utilized at 25 km horizontal resolution during JJAS of the year 2022.

**Table 1. Details of domain configuration and physics options used in HGFM.**

| Physics | Description |
|---|---|
| Radiation | Rapid Radiative Transfer Model (RRTM) for both Shortwave and Longwave (Iacono et al., 2000; Clough et al., 2005) with Monte Carlo Independent Column Approximation (McICA) |
| Microphysics | Formulated grid-scale condensation and precipitation (Sundqvist et al., 1989; Zhao and Carr, 1997) |
| Convection | Aerosol aware and Mass flux based Simplified Arakawa-Schubert (SAS) shallow convection (Pan and Wu, 1995; Han and Pan, 2011; Arakawa and Wu, 2013; Han et al., 2017) |
| Planetary Boundary Layer (PBL) | Hybrid Eddy-Diffusivity Mass Flux vertical turbulent mixing scheme (Han and Pan, 2011; Han et al., 2016) |
| Gravity Wave Drag (GWD) | Mountain blocking (Alpert et al., 1988; Kim and Arakawa, 1995; Lott and Miller, 1997) and stationary convective-forced GWD (Chun and Baik, 1998) |



## 3 Results and Discussions

### 3.1 200 hPa Kinetic Energy Spectra

Before going into the details of model validation, the first metric to evaluate the model fidelity is to validate the Kinetic Energy (KE) spectra of 200 hPa wind. The KE spectra provide information about the distribution of kinetic energy across the scale. A close resemblance between observed /reanalysis-based spectra and spectra produced by the model gives confidence about accuracy of overall model configuration. The kinetic energy (KE) spectrum in the upper troposphere exhibits two clearly defined power-law patterns. From observational studies, it is established that at large-scale, rotational modes prevail ($k^{-3}$) while at mesoscales, divergent modes are dominant ($k^{-5/3}$) (Nastrom and Gage, 1985). Figure 2 shows the KE spectra of 200 hPa wind simulated by HGFM and GFS T1534. The KE spectra for the forecast up to 3 days lead time has been compared with ERA5 data. While both the models reasonably capture $k^{-5/3}$ behaviour of the mesoscale at the higher wavenumber, but the HGFM appears to capture the $k^{-3}$ behaviour of the large scale at the lower wavenumber closer to observation. The KE spectra indicates that overall configuration of both versions of the model is robust. Therefore, now we turn our attention towards verification of convective available potential energy and rainfall simulations, the most desirable parameter in model forecasts.

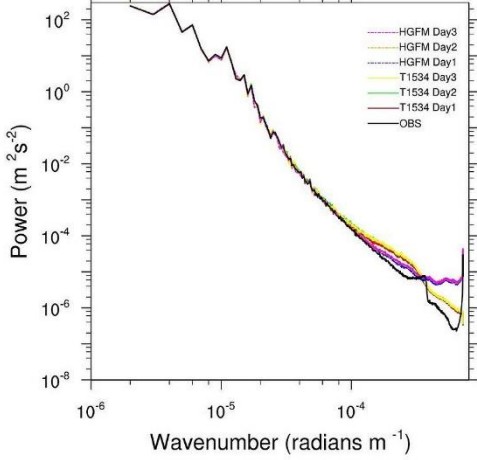

**Figure 2. Kinetic energy spectra of 200 hPa wind for observation and different lead times of GFS T1534 and HGFM.**

### 3.2 Quasi-equilibrium in models

Both model versions are run at high-resolutions, close to convection-permitting models' resolution. However, in this case, a scale-aware convection scheme is used to parameterize deep convection in the model. From observational studies it has been established that tropical atmosphere deviates significantly from the convective-quasi equilibrium (e.g., Zhang, 2003). The convective quasi-equilibrium (CQE) is the fundamental approach used in most models for parameterization of deep convection (Arakawa and Schubert 1974). To understand up to what extent both model versions obey CQE, we adopted



methodology suggested in Kumar et al. (2022). The absolute value of changes in Convective Available Potential Energy
(CAPE) at daily timescales is analysed from GFS T1534 and HGFM models for the year 2022 during JJAS and compared
with the ERA-5 data (Fig. 3). Notable changes were observed in the daily dCAPE values between GFS T1534 and HGFM
compared to ERA-5. The daily dCAPE values from ERA-5 (Fig. 3a, d) data matches better with the HGFM (Fig. 3e, f) than
GFS T1534 (Fig. 3b, c) for day 1 and day 3 lead times.

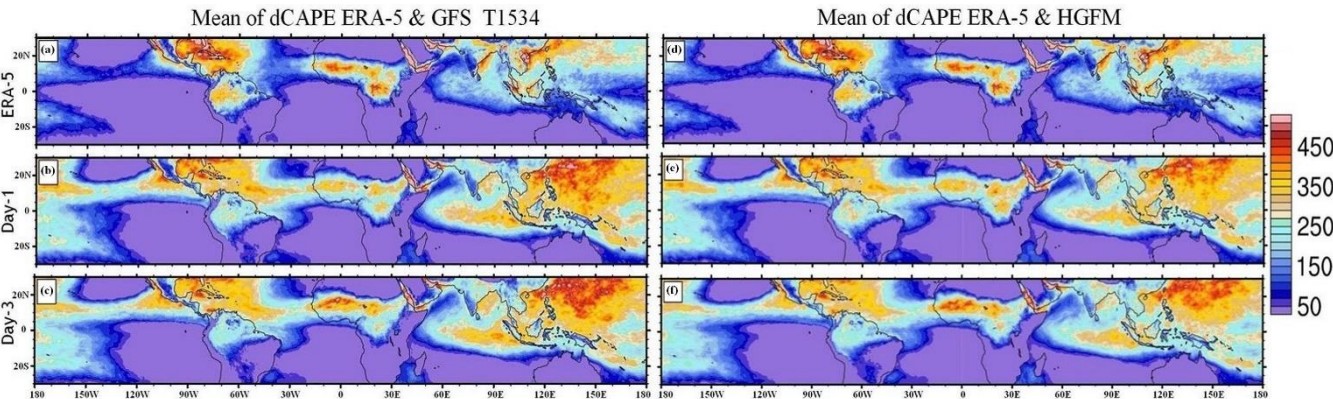

**Figure 3. Comparison of dCAPE mean during JJAS 2022 from ERA-5 (a, d) with respect to GFS T1534 (b, c) and TCO 1534 (e, f)**
**for day-1 and day-3 lead time.**
**3.3 Analysis of Global precipitation**
The global precipitation bias of GFS (left panel of Fig. 4 and HGFM (right panel) with respect to Integrated Multi-satellite
Retrievals for GPM (IMERG) data, with day 1, day 3 and day 5 lead time is shown in Fig. 4. Both the models broadly show
a similar rainfall bias over the global land and global ocean. However, there are some subtle differences. The day 1 forecast
(Fig. 4a) of GFS shows a wet bias over the equatorial eastern Pacific extending up to the tropical western Pacific. On the
other hand, the HGFM on day 1 lead (Fig. 4d) also shows a wet bias mostly confined over the tropical eastern Pacific and a
slight negative bias over western Pacific. For HGFM, the positive bias of rainfall over the tropical ocean appears to be
mostly over the eastern Pacific while that of GFS appears to be over eastern Pacific and extending towards the central and
west Pacific for all the lead time. Apart from the oceanic region, the major global land regions (central African Continent,
Maritime continent, Indian summer monsoon region, northern part of South America) shows a negative bias in both the
models at different lead times (Fig. 4) which is likely related to the model physical parameterizations.



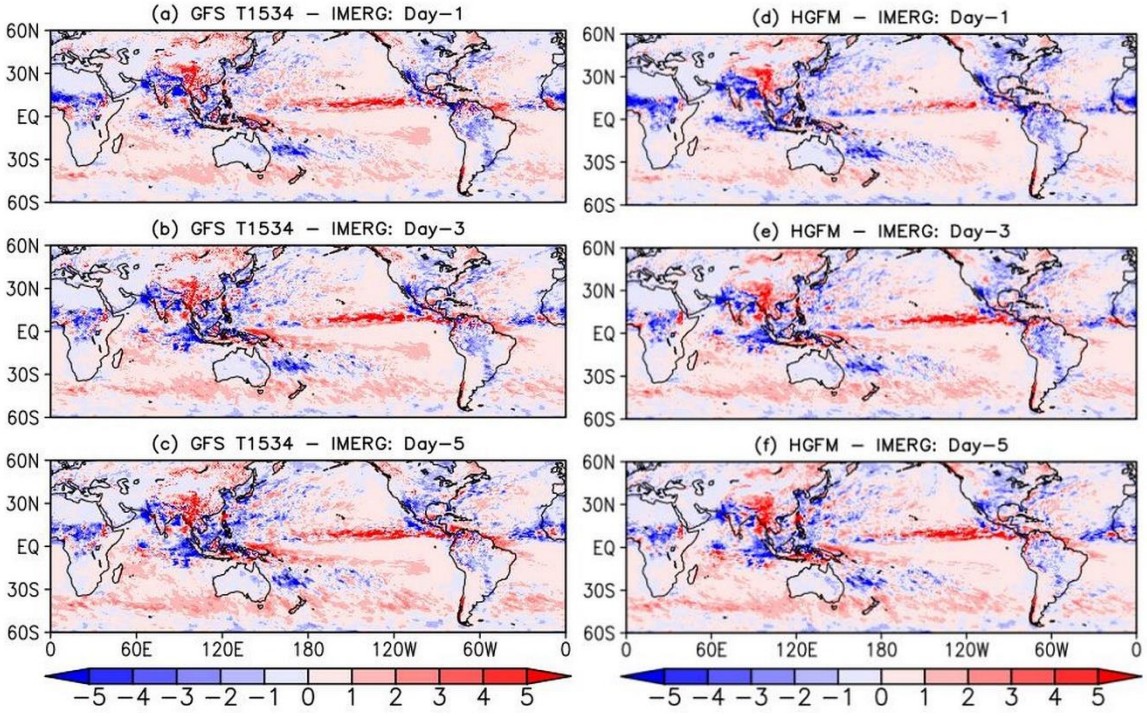

**Figure 4. Global JJAS precipitation bias (mm day-1) of GFS T1534 (left panel) with respect to IMERG for (a) day-1, (b) day-3 and (c) day-5 lead time. Right column (d-f) indicates similar plots but for HGFM.**

### 3.4 Indian summer monsoon precipitation and related features

While Fig. 4 depicted the precipitation bias over the global domain, it will be interesting to investigate the model forecast performance over the complex orographic region over the Indian domain, the region of our utmost interest. As mentioned earlier, one of the major advantages of using a Tco grid is a better representation of orography. Therefore, it is imperative to investigate the forecast skill of the high resolution HGFM model over the mountainous Himalayan foothills, adjoining northeast India, and Western Ghats (WGs) region (shown in Fig. 5 and 6 respectively). The GFS T1534 model forecasts indicate spurious rainfall activity over the Himalayan foothills and northeast India region for all lead times (Fig. 5b-d). On contrary, the HGFM model with finer horizontal resolution largely resolves the spurious rainfall over the region as shown in Fig. 5e-g. The Gibbs waves are largely suppressed over the mountainous terrains in HGFM compared to GFS T1534. Similarly, the precipitation distribution over the WGs region shows considerable overestimation in GFS T1534 for all lead times (Fig. 6b-d). On the other hand, the magnitude of overestimation is decreased considerably in HGFM forecasts as depicted in Fig. 6e-g. Thus, the above analysis brings out the fact that HGFM shows its potential in predicting realistic rainfall distribution over the orographic regions.



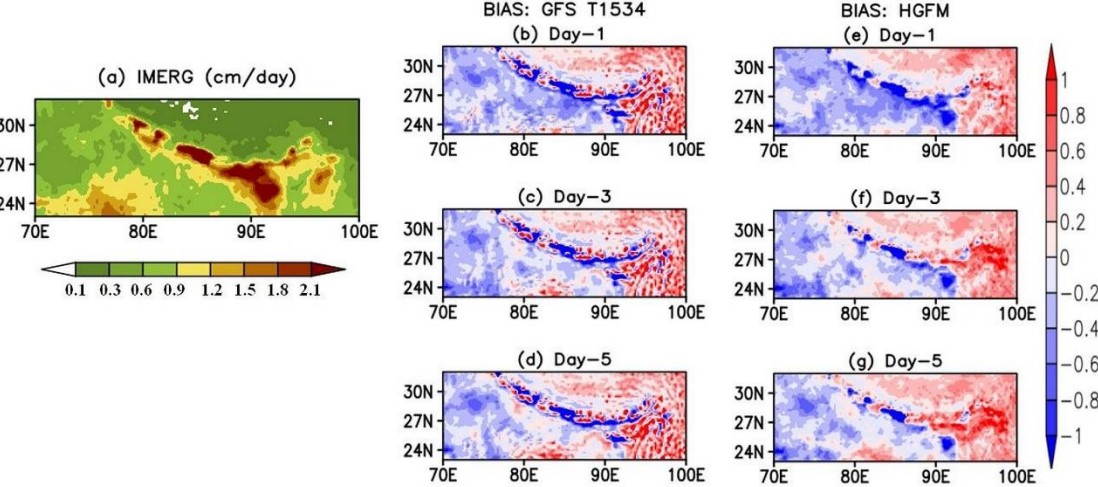

**Figure 5.** Comparison of JJAS mean precipitation (mm/day) and Bias in IMERG data (cm/day) (a) with GFS T1534 (b, c, d) and TCO 1534 (e, f, g) during 2022 over Himalayan foothills and Northeast India for day-1 day-3 and day-5 lead time.

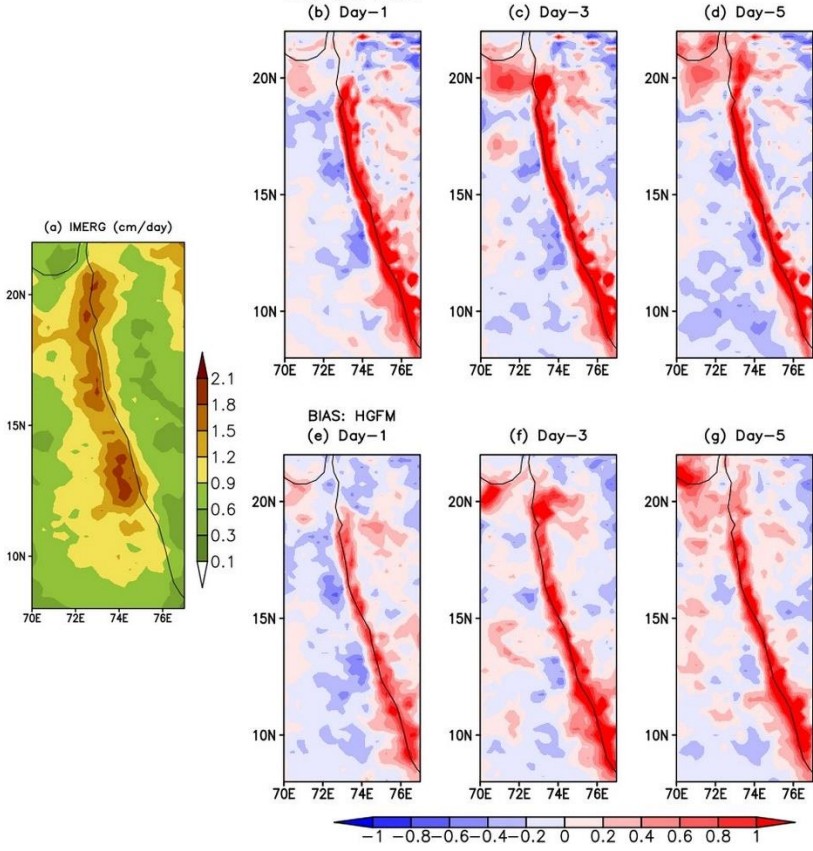

**Figure 6.** Comparison of JJAS mean precipitation (mm/day) and Bias in IMERG data (cm/day) (a) with GFS T1534 (b, c, d) and TCO 1534 (e, f, g) during 2022 over Western ghats region for day-1 day-3 and day-5 lead time.



One of the prominent features of ISM is vertical shear of zonal wind. Previous studies (Jiang et al., 2004; Abhik et al., 2013)
demonstrated that the vertical easterly wind shear plays a crucial role in inducing baroclinic vorticity ahead of northward
propagation of summer intra-seasonal oscillation. In order to find out the model forecast skill in predicting realistic easterly
wind shear (difference between zonal wind at 200 and 850 hPa) during summer monsoon season of 2022, the vertical wind
shear calculated and represented in Fig. 7a and 7b for GFS T1534 and HGFM respectively over the ISM region. Figure 7a
indicates slightly weaker easterly shear in GFS T1534 compared to ERA5 around $10^{o}$ N and $0^{o}$-$15^{o}$ S for all lead times. On
the contrary, the HGFM is able to predict more realistic easterly wind shear over above regions as shown in the Fig. 7b. It is
noticeable that both models overestimate the magnitude of easterly shear around $20^{o}$ N for Day-3 and Day-5 lead times.
Another key feature about tropical precipitation is almost equipartition of rainfall into convective and stratiform rain.
Therefore, it is important to investigate whether the relative improvement in the precipitation distribution over the ISM
region in HGFM forecasts is contributed by improved convective and large-scale precipitation. The model forecasted
convective and large-scale rainfall ratios are shown in Fig. 7c and 7d respectively. It is noteworthy that the large-scale or
stratiform rainfall plays an important role in the propagation and maintenance of the tropical intraseasonal convection
associated with its top-heavy heating profile (Fu and Wang, 2004; Chattopadhyay et al., 2009; Deng et al., 2015). The
heating profile associated with stratiform rain also helps in large-scale organization of convection (see for example,
Choudhary and Krishnan, 2011, Kumar et al., 2017). The contribution of convective rainfall to the total rainfall appears to be
more than 80 % in GFS T1534 forecast for all lead times (Fig. 7c). Similar overestimation of convective rainfall in GFS
T1534 is reported by Ganai et al. (2021). The observed convective (large-scale) rainfall ratio is around 55 % (45 %) as
shown in Abhik et al. (2017). The HGFM forecast shows relative improvement in predicting convective and large-scale
rainfall ratio compared to GFS T1534 (Fig. 7c and 7d). The decrease (increase) in convective (large-scale) rainfall
contribution to total rain is noted in HGFM forecast. The finer horizontal resolution in HGFM possibly allows for a more
accurate representation of deep convective due to scale-aware representation.



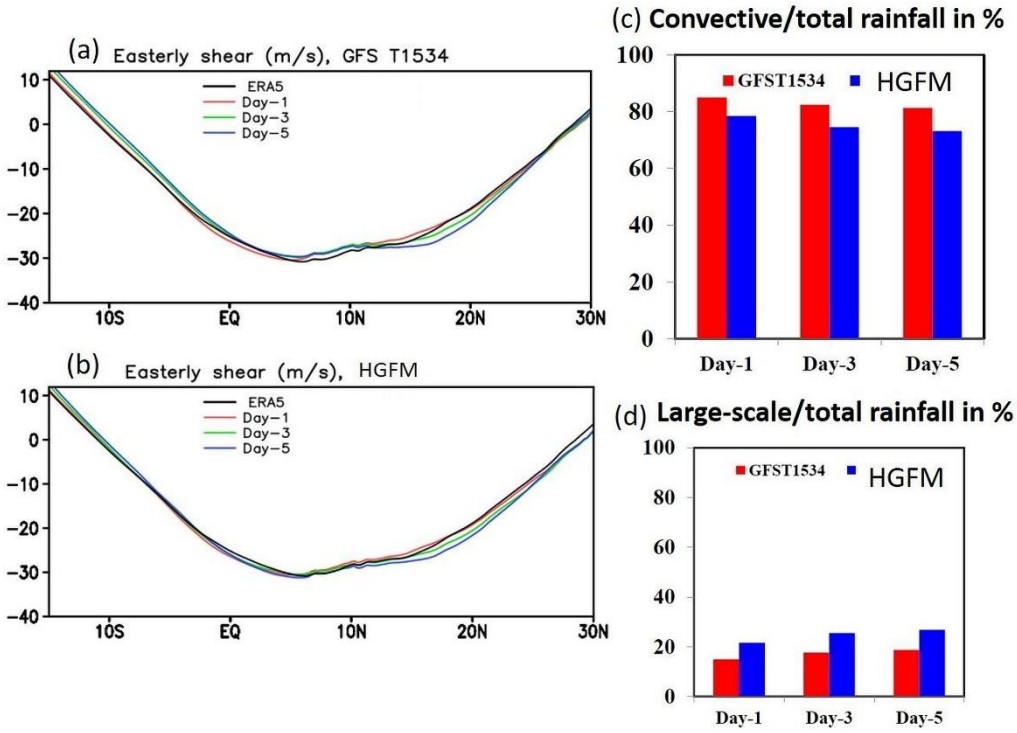


**Figure 7. Comparison of easterly shear (m/s) from ERA-5 with GFS T1534 (a) and HGFM (b) along with convective/total rainfall (c) and large scale/total rainfall (d) between GFS T1534 and HGFM during JJAS 2022 for day-1 day-3 and day-5 lead time.**

To attain further clarity about the model precipitation and moist convective processes, the vertical profile of relative humidity as a function of rain rate is analyzed for JJAS of 2022 over the ISM region ($60^{o}$ E-$100^{o}$ E, $10^{o}$ S-$30^{o}$ N). The bias analysis suggests that GFS T1534 has systematically underestimated the lower-level moisture for all lead times (Fig. 8b). It is consistent with the study by Mukhopadhyay et al. (2019) and Ganai et al. (2021) where they reported similar underestimation of lower-level moisture over the ISM region IN GFS T1534 forecast. In contrast, the HGFM shows relative improvement in the lower-level moisture distribution, as depicted in Fig. 4c for all lead times. The enhancement of the lower-level moisture is visible as compared to GFS T1534 forecast. However, the upper troposphere is too moist for both model forecasts and need further improvement.

It is observed that overall statistics of monsoon rainfall and related convective processes have significantly improved in the HGFM model. In the next section some recent tropical cyclone forecasts are analysed.



261

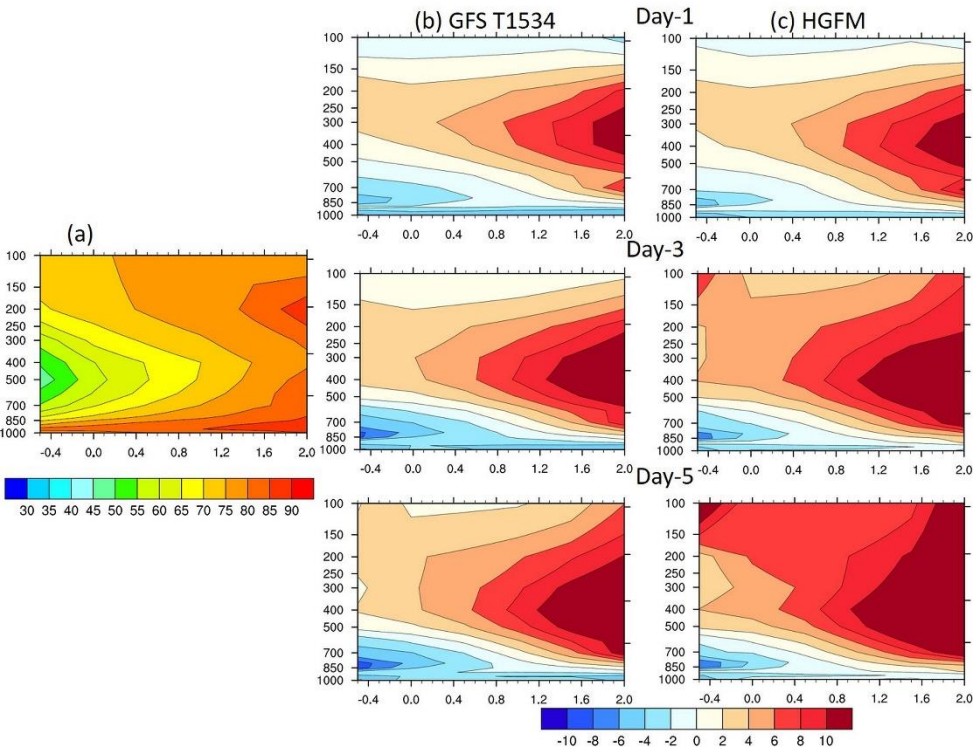

**Figure 8. Comparison of Relative humidity (%, bias in shaded) vs rain rate (mm/day) over ISM region (60° E-100° E, 10° S-30° N) during JJAS-2022 from ERA-5 and IMERG (a) with GFS T1534 (b) and HGFM (c) during JJAS 2022 for day-1 day-3 and day-5 lead time.**

**3.5 Evaluation of Heavy Rainfall event**

A very heavy rainfall event occurred on 22 August 2022 over central India. This event was well captured by both GFS and HGFM models as compared to the observed rain from IMD-GPM (shown in Fig. 9). Both HGFM (Fig. 9a, b, c) and GFS T1534 (Fig. 9d, e, f) models simulated the heavy rainfall signature compared to IMD GPM (Fig. 9g) on day 1 and day 3 forecast. However, a major difference was noted for rainfall intensity and spatial distribution on longer lead time (day 5) in HGFM and GFS T1534. There is an underestimation of rainfall in both the models compared to observations. Whereas the HGFM captures the signal of the occurrence of heavy rainfall even at day 5 lead, which is almost negligible in GFS forecast.

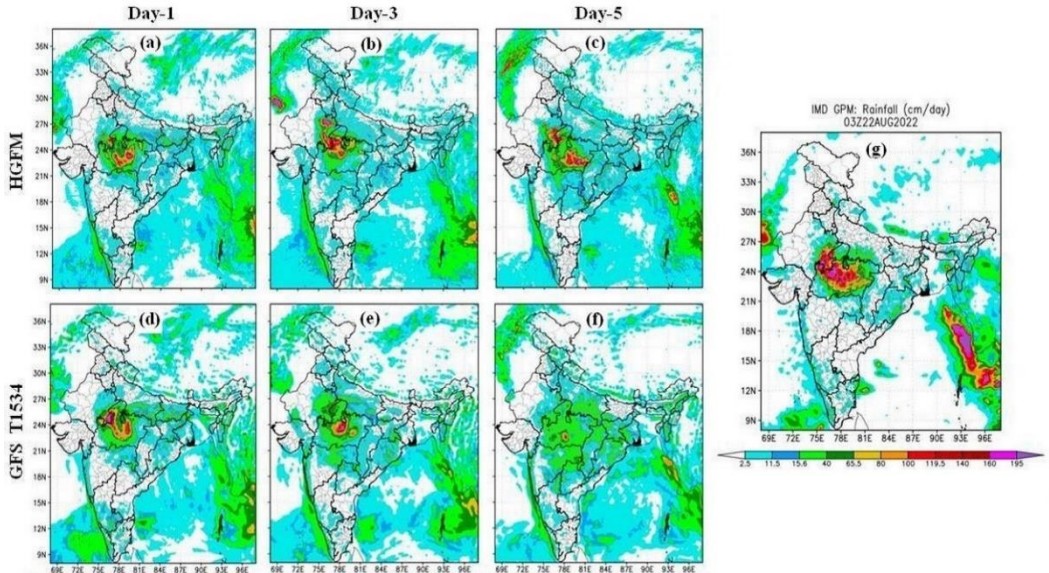

**Figure 9. Comparison of heavy rainfall event on 22 August 2022 with HGFM (a, b, c), GFS T1534 (d, e, f) for day-1, day-3 and day-5 lead times with IMD GPM (g) rainfall.**

## 3.5 Evaluation of Tropical Cyclone forecast

Total eight cases of tropical cyclones from 2022 and 2023 (RSMC 2022, RSMC 2023) are considered in the present study. Out of these 8 cases, 2 cyclones formed over the Arabian Sea and 6 cyclones over the Bay of Bengal (BOB). The observational data of track, intensity and landfall is obtained from IMD and referred as observations henceforth in the text. Figure 10 shows observed tracks (Fig. 10a) and observed intensity in terms of Maximum Sustained Wind Speed (MSW Fig. 10b) of the cyclones. The cyclones in the present study have different tracks and various range of severity in terms of intensity over both the basins.

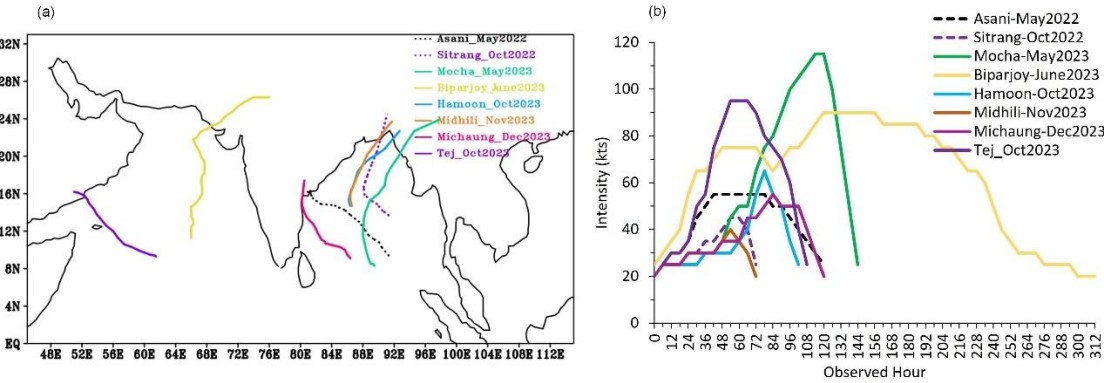

**Figure 10. a) Observed tracks of the cyclones b) Observed Intensity in terms of Maximum Sustained Wind Speed (kts) during year 2022-2023.**





### 3.5.1 Annual Verification of GFS T1534 and HGFM Forecast for the year 2022 and 2023

For each cyclone case, the verification started from the observed Depression stage till observed landfall. For each cyclone case minimum four (maximum 10) initial conditions are considered as both the models have daily outputs. The errors calculated here are the average of all such samples for the year 2022 and 2023.

The Root Mean Square Error (RMSE) of track and intensity is shown in Fig. 11a-b. Initially upto 4 days, GFS T1534 and HGFM performs equally well but the considerable improvement with HGFM is noted after 4 days in both track and intensity forecast. Figure 11c-d   depicts the average track error and average intensity errors for all the cyclones. The average track errors as well as average intensity errors are reduced drastically in HGFM with longer lead hours (4 days or more). Average track errors (average intensity errors) are ~300 km (~20 kts) with 7 days leads in HGFM. The average landfall errors (both position and time) are also evaluated with IMD observations and are shown in Fig. 12. With 4days lead, average landfall position errors are ~200 km in HGFM and it reduces further with longer lead. In GFS T1534, landfall position errors are increasing with longer lead (compared to HGFM). Remarkable improvements are seen in the average landfall time errors in HGFM throughout the life cycle of cyclones. Overall, the track and intensity forecast are improved with HGFM for longer lead hours (~4 days or more), which is an added advantage for the early warning and mitigation purpose. Here, one of the cyclone cases (cyclone Biparjoy) is discussed in detail.

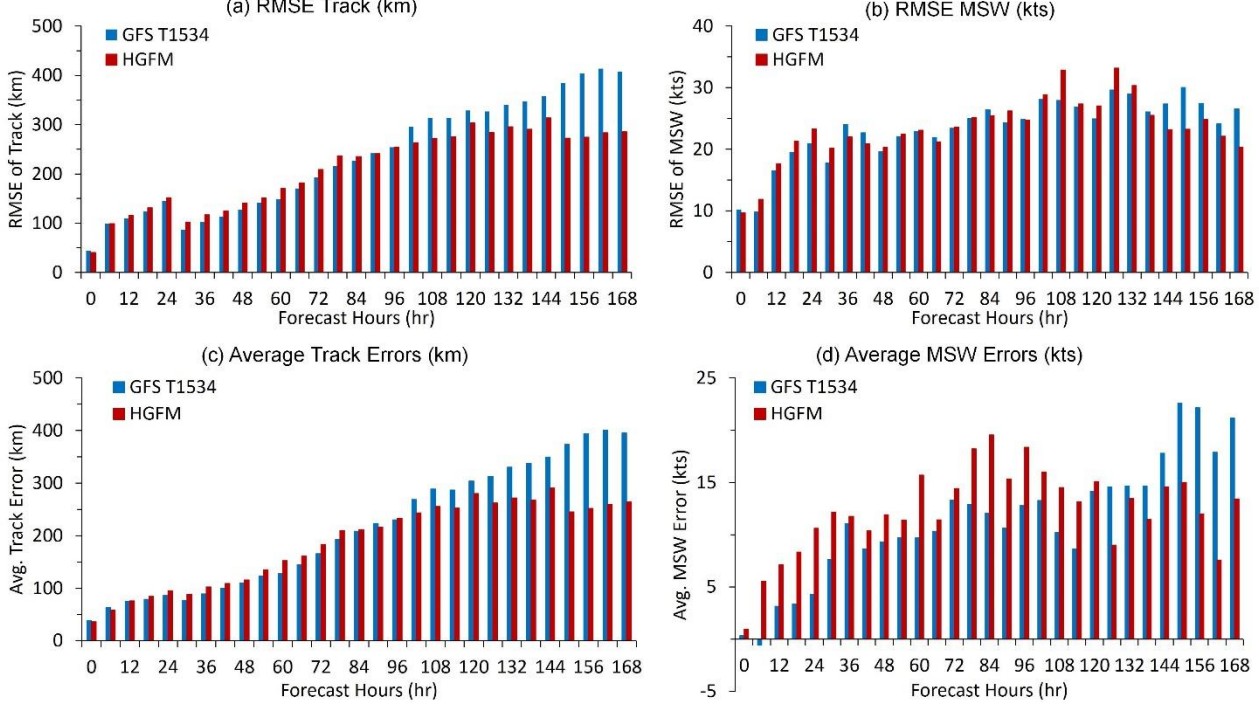

**Figure 11. a) RMSE of Track in km b) RMSE of MSW in kts c) Average Track error (km) d) Average Intensity Errors (kts).**





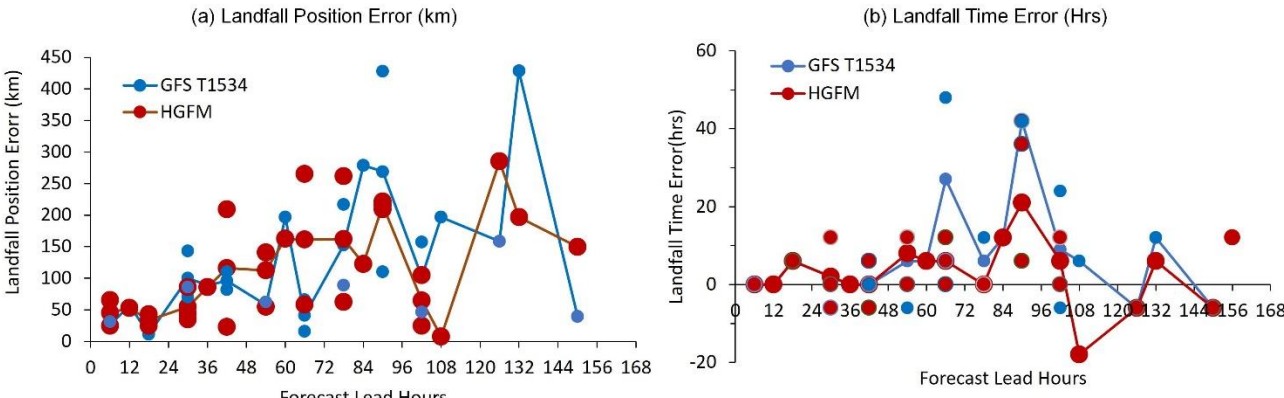

**Figure 12. a) Average Landfall position errors in km b) Average Landfall time Errors in hours. The continuous lines represent the average errors for GFS T1534 (Blue) and HGFM (Red). The different size of the dots is for making the overlapped points visible.**

**3.5.2 A case study - Cyclone Biparjoy**

During the monsoon onset of 2023 season, tropical cyclone Biparjoy evolved in the Arabian Sea and hit the north-western state of Gujarat, India. The cyclone Biparjoy lasted for quite a long time during 6-19 June 2023. It moved almost parallel to the Indian west coast and finally made landfall over the northern part of Gujarat and adjoining Pakistan. It has rapid intensification during its life cycle. The observed track shown in Fig. 13 as provided by IMD.

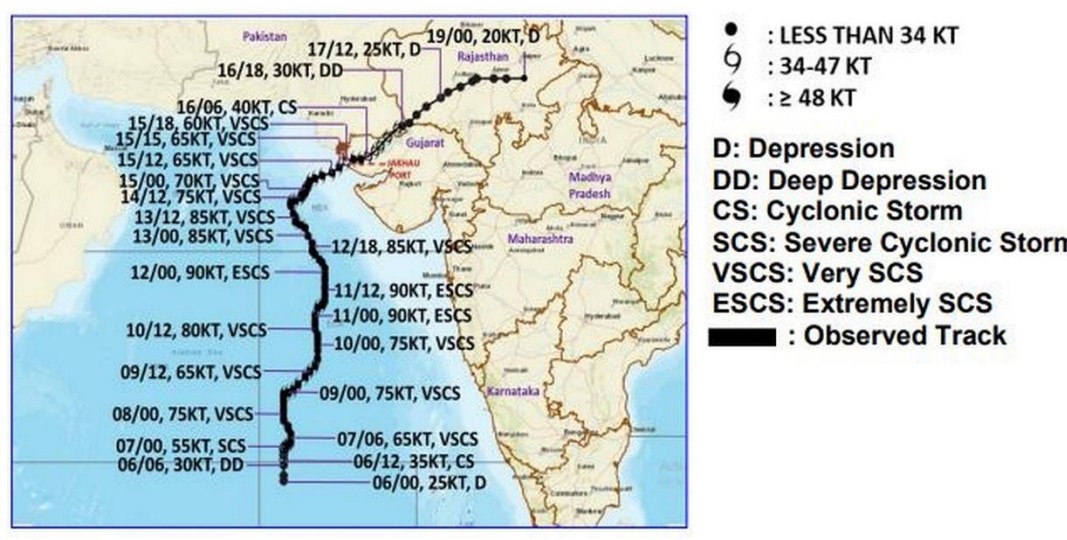

**Figure 13. Observed track of Tropical cyclone Biparjoy over Arabian Sea during 6-19 June 2023 as per India Meteorological Department. Taken from open source: https://rsmcnewdelhi.imd.gov.in/report.php?internal_menu=Mjc= , page no. 90**





The HGFM and GFS T1534 track forecast of TC Biparjoy based on 6 June initial condition, is shown in Fig. 14. It is evident
that the HGFM generates a track much closer to the observation as compared to GFS T1534. The intensity expressed in
terms of maximum sustained wind has been computed and shown in Fig. 13 for 10 days (240 forecast hours). The intensity
of the TC appears to be overestimated by both the models till 120 hrs of forecast and thereafter the intensity seems to be
reasonably predicted with 6 June 0000 UTC initial condition. Both the models are tested with different initial conditions
(from 6 June 00UTC to 15 June 00UTC, every 24 hrs). A comparative analysis of landfall position and landfall time errors
with HGFM and GFS T1534 with respect to the observations obtained from IMD has been mentioned in Table 2. It is
evident that the landfall position error of the cyclone has been significantly improved by HGFM forecast though the landfall
time error appears to be almost equivalent as compared to GFS T1534. Further the average track and intensity error
(obtained from 10 initial conditions) is depicted in Fig. 14a and 14b. It is evident that the HGFM produces more accurate
prediction of track with lesser error on longer lead while the errors are equivalent in the smaller lead.
**Table 2. Landfall position (km) and landfall time (hr) errors for the forecasts started with different initial conditions. -ve (+ve )**
**sign indicates early (late) landfall with respect to observed landfall time. The bold numbers indicates the significant improvement**
**in the landfall position errors with HGFM.**

| Forecast Hours from Observed landfall (Hr) | Initial Condition | Landfall Position Error (km) | | Landfall Time Error (Hr) | |
|---|---|---|---|---|---|
| | | GFS T1534 | HGFM | GFS T1534 | HGFM |
| 228 | 2023060600 | 298 | **57** | 0 | -30 |
| 204 | 2023060700 | No Landfall | | | |
| 180 | 2023060800 | 616 | **201** | 0 | 0 |
| 156 | 2023060900 | 349 | **197** | 12 | 12 |
| 132 | 2023061000 | 428 | **197** | 12 | 6 |
| 108 | 2023061100 | 197 | **7** | 6 | -18 |
| 84 | 2023061200 | 279 | **123** | 12 | 12 |
| 60 | 2023061300 | 197 | **163** | 6 | 6 |
| 36 | 2023061400 | 89 | **86** | 0 | 0 |
| 12 | 2023061500 | 57 | **53** | 0 | 0 |




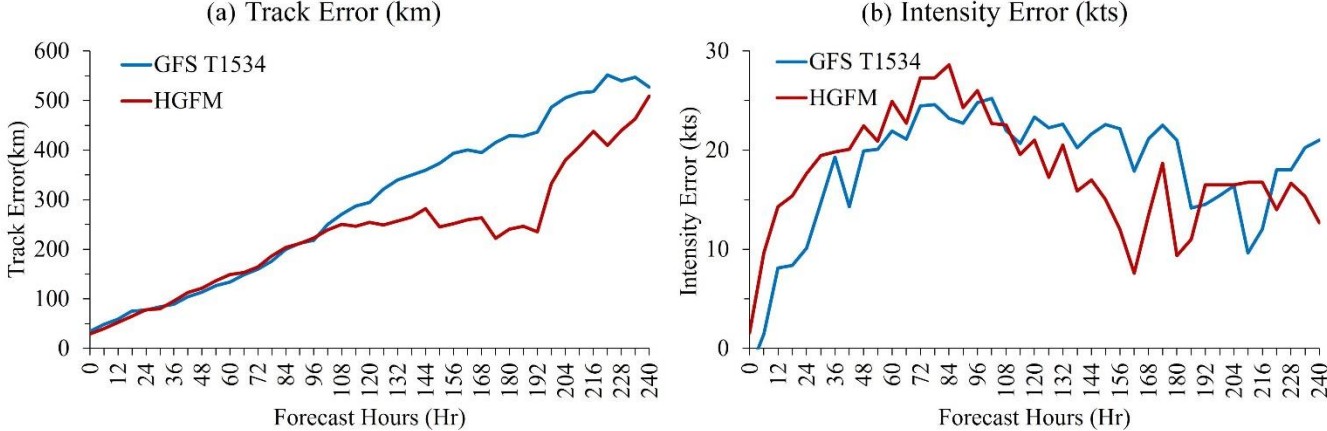


**Figure 14. a) Average track error and b) average intensity error for the tropical cyclone Biparjoy over Arabian Sea.**

## 4 Conclusions

For the first time, a version of the GFS model utilizing a new grid structure triangular cubic octahedral (Tco) has been
developed and is being run on an experimental basis for short to medium range weather prediction over the Indian region,
designated as IITM High resolution Global Forecast Model (HGFM). The Tco grid provides a higher resolution over the
tropics, making the model achieve 6.5 km horizontal resolution near the tropics. This higher resolution represents a
substantial leap from the existing Gaussian linear GFS T1534 which maintains a resolution of 12.5 km across the globe. The
KE spectra of 200 hPa zonal wind have also revealed reasonable power by both the model with HGFM showing marginally
better power in the Kolmogorov region indicating fidelity of model structure.
The HGFM being developed in the Tco grid provides many advantages, notably resolving the Gibbs phenomenon and
spurious rain over mountainous regions has been resolved. The June-September monsoon rainfall and a case study of heavy
rainfall have been analyzed in detail. The newly developed HGFM shows significantly better skill, particularly in the longer
lead and for heavier rain categories. Rainfall biases over the whole globe appear to be broadly similar between HGFM and
GFS T1534. A case of heavier rainfall in and around central India during the monsoon season has been analysed where the
validation shows a significant gain in forecast lead time by the HGFM compared to GFS T1534. The HGFM captures
rainfall signature at 5 days lead time, when there is hardly any indication in the HGFM model forecast.
Several cases of tropical cyclones during 2022 and 2023 were analysed, indicating better performance of HGFM compared
to GFS in predicting tracks and intensity. A case of tropical cyclone Biparjoy has been evaluated in detail based on IMD
observation. It is seen that the HGFM model generates better accuracy of cyclone position in almost all lead time (Table 2)
and further the average track error also is found to be much lesser as compared to GFS T1534 in longer lead. However, the
errors of both model in average track and intensity are found to be equivalent.





This paper highlights the initial results of the newly developed HGFM model and its skill as compared to the operational
GFS T1534 model. Subsequently more analyses for many events will be carried out and the model will be made operational
for weather forecasts over India. The current set up of the model uses the same physics as the GFS model. However, the
HGFM model would require some parameter tuning to optimize the performance of the model and increase its fidelity. The
future work will be focused on detailed validation of model simulations with optimal set of physical parameterizations.

**Code and Data Availability**
The model simulated data used for HGFM and GFS T1534 in the study are available at "TCO model data" by R Phani
Murali Krishna, Kumar Siddharth, Athipatta Gopinathan Prajeesh, Malay Ganai, Revanth Reddy, Kumar Roy and
Parthasarathi Mukhopadhyay, DOI: https://doi.org/10.5281/zenodo.12569807. The model code is available at "GFS TCO
Model code" by R Phani Murali Krishna, Kumar Siddharth, Athipatta Gopinathan Prajeesh, Parthasarathi Mukhopadhyay.
DOI: https://doi.org/10.5281/zenodo.12526400

**Author Contributions**
RPMK, SK, AGP and PM conceptualised the problem and made necessary changes/modification development of code for
Tco and wrote the major part of the Introduction, data, methodology and over all sequences. PB and NW helped during
formulation of the Tco grid in GFS and helped in improving the manuscript writing. KR, MG, ST and TG made all the
forecast analysis of monsoon parameters and wrote the respective portion on analyses. RK made the analysis related to
cyclone forecast by HGFM model and wrote the section on the cyclone forecast analysis and BRR made the dCAPE analysis
and extracted the post processed variables for the analysis.



**Competing interests**
The authors declare that they have no conflict of interest.


**Disclaimer**







**Acknowledgments**
IITM is fully funded by the Ministry of Earth Sciences, Government of India. We would like to thank ECMWF for their
support during the model development and for providing the ERA5 data set. We thank NCMRWF for providing the GFS
initial conditions used for conducting simulations. We acknowledge Pratyush High Performance Computing at IITM, Pune
for providing the computing facility to carry out the simulations. We thank Mr. Vaishak for helping in archiving the data in
ARDC server. Authors thank Secretary Ministry of Earth Sciences, Government of India and Director, IITM for support and
facilities provided for this study. We thank IMD for providing the GPM rainfall and cyclone best track data.















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
