# Peer review of "IITM High-Resolution Global Forecast Model Version 1: An attempt"

_Geoscientific Model Development, 2024_

## Author Comment (AC1)

**Referee #1**

The authors introduced a new model that can significantly improve the prediction of monsoon, which is key to local agriculture, economy, and disaster preparedness. The prediction ability of this new model is encouraging. I think the manuscript is clearly written although some loose ends need to be addressed. Specifically, I found that the overall quality of the figures varies a lot. High-quality figures are important to convey key results. I think a major revision is needed to address this. Then it could be published at GMD. Please see my detailed comments below.

Reply: We thank the reviewer for the constructive comments and suggestions. The figures have been added with high-quality and addressed all the reviewer comments below...

1. Figure 1c, the legend needs to be fixed (solid lines vs. dash lines)

   Reply: We thank the reviewer for the suggestion. The below figure shows the legend modification with the same single y-axis scale.

[Figure]

   FIGURE R1: Variation of grid length with latitude in GFS (blue) and Tco (red) (a), depiction of grid resolution over the globe in Tco grid (b), total and dynamics time taken for different number of cores (c). Time taken by GFS and HGFM for one day forecast (Left vertical axis is total time taken and model dynamics time multiplied by 3).

2. Also figure 1c, it is inconvenient to compare the ratio of dynamics time to total time. Perhaps only using one single y-axis?

   Reply: In Figure R1, single Y-axis is used to compare the total and dynamics time (multiplied by 3).

3. Why only considering 200 hPa kinetic energy? How about other vertical levels?

   Reply: Thank you for the valuable comment. As maximum kinetic energy and potential to kinetic energy conversion occurs in the upper troposphere, we display kinetic energy spectra at the 200 hPa level.

4. Figure 3: It would be better to add the model names to each panel. I am confused by lines 188-189. To me, HGFM and GFS look more similar to each other while ERA5 look quite different, especially over the gulf of Mexico and northwestern Pacific. I am suggesting plotting the difference for better comparisons among the three outputs.

Reply: Thank you for the suggestion. The Difference of dCAPE between ERA-5 and models is presented for day-1 & day-3 lead time forecasts. The dCAPE differences quantified from ERA-5 with GFS T1534 were –49.0570 (J/kg/day) and –47.3799 (J/kg/day) for day-1 and day-3 lead times respectively, similarly with HGFM – 49.1278 (J/kg/day) and –43.7668 (J/kg/day) for day-1 and day-3 lead times respectively, the quantified values will be included in the manuscript.

[Figure]

FIGURE R3: The difference of dCAPE from ERA-5 and GFS T1534 for day-1 and day-3 (left panels), and from ERA-5 and HGFM for day-1 and day-3 (right panels).

5. Figure 4 and line 194 to 203: there are some discussions about the biases over the tropical ocean. I am wondering if there are any specific reasons why both models overestimate the precipitation over the tropical eastern Pacific? Is it due to the shallow convection scheme?

Reply: Most of the CMIP5 models overestimate precipitation over the tropical eastern pacific. Precipitation biases over tropical oceans are largely dependent on model physics, i.e. convection and cloud radiation interaction, and show little dependence to model resolution.

6. Figure 5&6: is it cm/day or mm/day?

Reply: Thank you for the suggestion. It is cm/day and modified in the revised version.

7. Figure 7: improve the quality of this figure (mixed font sizes, panel sizes, etc.).

Reply: Thank you for the suggestion. It is corrected and modified.

8. Why convective precipitation is reduced from GFS to HGFM? Due to Tuning?

Reply: Higher resolution possibly helps better resolving the topography etc. and resolves mesoscale convection which is manifested through improved large scale precipitation (Fig. 7d) and reduced sub-grid scale precipitation (Fig. 7c). The convection in this model uses the scale aware scheme of Han et al. (2017) where the scheme adjusts the proportion of sub-grid scale convection and grid-scale which appears to be more effective in HGFM (being a variable grid model) than the Gaussian linear GFS T1534.

9. What is the point of figure 13?

Reply: This figure is added to provide the official track provided by India Meteorological Department.

---

## Author Comment (AC2)

**Referee #2**

The manuscript proposes an improvement to the GFS numerical prediction model by employing a dynamic core based on a cubic octahedral grid, enabling an increase in model resolution to 6.5 km over the tropics. The model is then applied to resolve monsoon convection, producing daily runs from June to September 2022. The results are compared to those obtained using the operational GFS T1534. The authors find that their modeling approach performs significantly better, particularly for longer lead times and heavier rain events.

This is an interesting paper, as the suggested method appears to improve weather forecasts in regions with intense precipitation. Its content aligns well with the scope of Geoscientific Model Development (GMD) journal. However, I believe several improvements are necessary before the manuscript can be considered for publication.

Reply: We thank the reviewer for the constructive comments and suggestions. We have addressed all the comments below.

General Comments

1. Reference to Previous Work

   Based on a web search, it seems that enhancing the resolution of numerical weather prediction models is not a novel concept and has already been applied (e.g., https://doi.org/10.1002/qj.958). However, such references are missing from the manuscript, and I suggest that the authors update the reference list accordingly.

Reply: Thank you for the suggestion. The new reference (Staniforth and Thuburn, 2012) will be added in the manuscript as suggested by the reviewer. Some of the references (Satoh et al., 2005; Miura et al., 2007; Satoh et al., 2019) on the enhancing the model resolution will be included in the manuscript. A new reference on the ICOsahedral Non-hydrostatic (ICON) model (Majewski et al., 2002) and a study on the precipitation over East Asia with different model resolutions by Li et al., 2015 also be added in the manuscript which was found relevant for this study.

Staniforth, A. and Thuburn, J.: Horizontal grids for global weather and climate prediction models: a review. Q. J. R. Meteorol. Soc., 138(662), 1-26, https://doi.org/10.1002/qj.958, 2012.

Majewski, D., Liermann, D., Prohl, P., Ritter, B., Buchhold, M., Hanisch, T., Paul, G., Wergen, W., and Baumgardner, J.: The operational global icosahedral-hexagonal gridpoint model GME: description and high resolution tests, Mon. Wea. Rev., 130, 319– 338, https://doi.org/10.1175/1520-0493(2002)130<0319:TOGIHG>2.0.CO;2, 2002.

Li, J., Yu, R., Yuan, W., Chen, H., Sun, W. and Zhang, Y.: Precipitation over E ast A sia simulated by NCAR CAM5 at different horizontal resolutions. J. Adv. Model. Earth. Syst., 7(2), 774-790, https://doi.org/10.1002/2014MS000414, 2015.

2. Clarification of Innovative Contribution

   Given the above observation, the innovative aspect of this work appears to be the application of a dynamic core using a cubic octahedral grid specifically within the GFS

model, rather than the general application of this method to numerical weather prediction models. If this is the case, it should be clearly stated in the text.

Reply: Thank you for the suggestion.

It is worth to mention that the present dynamical core using cubic octahedral grid is implemented in ECMWF weather forecast model since 2016 (Malardel et al. 2016). This has led to a significant increase in forecast accuracy and computational efficiency in the ECMWF model. In the present study, it is found that the above dynamical core in the GFS T1534 has improved the orographic rainfall and reduces the Gibbs noise over the mountainous region in addition to improved precipitation skill over the Indian landmass region.

This will be added in the "conclusion" part in the revised manuscript.

Malardel, S., N. Wedi, W. Deconinck, M. Diamantakis, C. Kühnlein, G. Mozdzynski, M. Hamrud & P. Smolarkiewicz, 2016: A new grid for the IFS. ECMWF Newsletter No. 146, 23–28.

3. Validation Period

The authors mention (lines 86–87) that the motivation for improving numerical weather forecasts in India under heavy precipitation conditions stems from difficulties in forecasting extreme rainfall over Kerala during August 2018 and August 2019. Therefore, it is unclear why they did not validate their approach for these events, opting instead for the June–September 2022 period. Additionally, limiting the validation to a single year raises concerns. It would be valuable to see how the model performs for another year or in different periods.

Reply: Thank you for the suggestion. We would like to thank the reviewer for the valuable comment. It is worth to mention that the present HGFM model is developed and made operational since 2022 monsoon season. The IC's before 2022 season is not available with us. Therefore, we are presenting only events which occurs during 2022 and later. One heavy rainfall event over central India during 22August 2022 is already included in the present study. However, as the reviewer wish to know about the model skill in predicting heavy to extreme rainfall event in HGFM. We have plotted the precipitation probability distribution function (PDF) over the Indian landmass region for 2023 summer monsoon. Although HGFM shows overestimation in the lighter category (0.25-1.56 cm/day) rainfall, it shows better PDF in the very heavy (11.56-20.45 cm/day) and extreme (>20.45 cm/day) rainfall category as compared to GFS T1534.

In the revised version of the manuscript, we will incorporate the result of rainfall PDF analysis (figure will not be shown) as per the reviewer's suggestion in section 3.5.

[Figure]

FIGURE. Precipitation probability distribution function (%) over the Indian landmass region during JJAS 2023 for (a) Day-1, (b) Day-3, and (c) Day-5 lead time based on IMERG (Black bar), GFS T1534 (Red bar) and HGFM (Blue bar).

4. Linguistic Improvements

The manuscript contains several syntax errors, which should be corrected, preferably by a native English speaker. Below are a few examples:

a. Line 108: "This A new grid" → "This new grid"

 Reply: Thank you for the suggestion. It was modified in the manuscript.

b. Line 146: "...Tco model is in general..." → "...the Tco model is in general..."

Reply: Thank you for the suggestion. It was modified in the manuscript.

c. Line 148: "...of the model run has been discussed..." → "...of the model run is discussed..."

Reply: Thank you for the suggestion. It was modified in the manuscript.

d. Line 149: "...few case studies have also been discussed" → "...few case studies are also discussed"

Reply: Thank you for the suggestion. It was modified in the manuscript.

e. Line 185: "...methodology suggested..." → "...the methodology suggested..."

Reply: Thank you for the suggestion. It was modified in the manuscript.

f. Line 226: "...is vertical..." → "...is the vertical..."

Reply: Thank you for the suggestion. It was modified in the manuscript.

Specific Comments

1. Lines 103–104

   The authors state:

   "This paper is the first attempt, to the best of our knowledge, towards building a model close to a convection-permitting global weather model in India, with an emphasis on Indian monsoon rainfall variability."

   It would be helpful if the authors clarified whether the improved performance of their model is expected to hold only for this region or if it could be effective in other geographical areas under similar conditions. If so, they should specify what these conditions might be.

Reply: Thank you for the suggestion. The HGFM model developed with TCO grid reduces the grid resolution from 12.5 km to 6.5 km near the tropics (Figure 1a). The study is mainly focused on the Indian region for testing the Indian summer monsoon which affect the billions of people's lifestyle. However, in Figure 4, the JJAS mean precipitation is depicted over the global region during 2022 JJAS. It is indeed showed slight improvement in the model biases over the global land regions compared to GFS T1534. The results are mentioned in the manuscript. In addition, the detailed global bias over different parts of the globe will be investigated in the future study.

2. Lines 154–160

   The authors compare forecasts with daily precipitation data from satellite products, gridded data from the India Meteorological Department (IMD), and the ERA5 database. However, given that previous studies highlight discrepancies between gridded data and ground measurements, it would be valuable to include a comparison with observations from ground-based meteorological stations. These are likely available in abundance for this region.

Reply: Thank you for the suggestion. The daily precipitation gridded data from IMERG include all the satellite and ground-based measurements data. The mean biases in the gridded data were corrected with IMD gauge data (Mitra et al., 2014). The ground based measurement at a point are highly specific and localized while the gridded data represents the average over entire grid and the model also produces a grid mean precipitation. In view of this, the model rainfall is compared with gridded observed rainfall.

3. Figure 2

The authors should discuss the discrepancies between the curves observed for wave numbers greater than 10−4 m−1 and whether these differences pose challenges to their analysis.

Reply: The regions of interest in KE spectra are the $k^{-3}$ dependence for the large scale and a less steep, $k^{-5/3}$ dependence for the mesoscale. The tail of the spectra at higher wave numbers typically has less energy due to the dissipation of kinetic energy with increase of wave number, however models tend to dissipate the energy at higher wave number at a much faster rate depending on the damping used in the model (Skamarock 2004). To keep the spectra realistic, one reduces the damping which may increase the energy at higher wavenumbers as observed for HGFM. However, this will not have much impact in our analysis as these are the small-scale features.

Skamarock, W.,C., (2004). Evaluating Mesoscale NWP Models Using Kinetic Energy Spectra. Monthly Weather Review,Vol. 132, p 3019–3032 ,DOI: https://doi.org/10.1175/MWR2830.1

4. Figure 3
   The differences between dCAPE values from ERA5 and HGFM should be quantified, as visual inspection alone may not suffice to identify discrepancies. Additionally, the units of dCAPE should be provided next to the color code for clarity.

Reply: Thank you for the suggestion. The Difference of dCAPE between ERA-5 and models is presented for day-1 & day-3 lead time forecasts. The dCAPE differences quantified from ERA-5 with GFS T1534 were –49.0570 (J/kg/day) and –47.3799 (J/kg/day) for day-1 and day-3 lead times respectively, similarly with HGFM –49.1278 (J/kg/day) and –43.7668 (J/kg/day) for day-1 and day-3 lead times respectively, the quantified values will be included in the manuscript.

[Figure]

FIGURE R3: The difference of dCAPE from ERA-5 and GFS T1534 for day-1 and day-3 (left panels), and from ERA-5 and HGFM for day-1 and day-3 (right panels).

---

## Author Response (AR2)

**Referee #1**

I would like to thank the authors for addressing my comments. However, I believe there is still room for improvement in the presentation of the figures throughout the manuscript. I recommend that the authors use consistent coding conventions when generating the figures.

Reply: We thank the reviewer for the valuable comments and suggestions.
The figures have been added with high quality and the necessary grammatical corrections were made throughout the manuscript.

**Referee #2**

Dear authors,

Thank you for addressing all the issues I raised. In my previous report, I suggested some indicative corrections related to English syntax. However you should have revised the complete manuscript. There is still room for improvement. This is why I suggest acceptance subject to technical corrections.

Reply: We thank the reviewer for the valuable comments and suggestions. We have made the necessary grammatical corrections throughout the manuscript.